# Childhood Obesity and Incorrect Body Posture: Impact on Physical Activity and the Therapeutic Role of Exercise

**DOI:** 10.3390/ijerph192416728

**Published:** 2022-12-13

**Authors:** Valeria Calcaterra, Luca Marin, Matteo Vandoni, Virginia Rossi, Agnese Pirazzi, Roberta Grazi, Pamela Patané, Giustino Simone Silvestro, Vittoria Carnevale Pellino, Ilaria Albanese, Valentina Fabiano, Massimiliano Febbi, Dario Silvestri, Gianvincenzo Zuccotti

**Affiliations:** 1Department of Internal Medicine, University of Pavia, 27100 Pavia, Italy; 2Pediatric Department, “Vittore Buzzi” Children’s Hospital, 20154 Milan, Italy; 3Laboratory of Adapted Motor Activity (LAMA), Department of Public Health, Experimental Medicine and Forensic Science, University of Pavia, 27100 Pavia, Italy; 4Department of Rehabilitation, Città di Pavia Hospital, 27100 Pavia, Italy; 5Research Department-LJA-2021, Asomi College of Sciences, 2080 Marsa, Malta; 6Department of Industrial Engineering, University of Tor Vergata, 00133 Rome, Italy; 7Department of Biomedical and Clinical Science, Università Degli Studi di Milano, 20157 Milan, Italy; 8Laboratory for Rehabilitation, Medicine and Sport (LARM), 00133 Rome, Italy

**Keywords:** obesity, body posture, children, adolescents, physical activity, exercise, musculoskeletal complication

## Abstract

Obesity is associated with various dysfunctions of the organism, including musculoskeletal problems. In this narrative review, we aim to consider postural problems in children and adolescents with obesity, focusing on the relationship with its negative impact on physical activity, and to discuss the role of exercise as a therapeutic approach. The body reacts to excess weight by changing its normal balance, and the somatosensory system of children with obesity is forced to make major adjustments to compensate for postural problems. These adaptations become more difficult and tiring if activities that require continuous postural changes and multi-tasking are engaged in. Children with obesity have less body control and functional ability due to the excess fat mass, which reduces their ability to perform motor skills and take part in physical activity. Appropriate early interventions for the management of musculoskeletal problems are needed to ensure healthy growth and to prevent comorbidities in childhood and adulthood. Prevention programs must be based not only on the reduction of body weight but also on the definition of correct postural habits from an early age. It is equally important to provide correct information on the types and doses of physical activity that can help prevent these problems.

## 1. Introduction

Obesity is defined as an excessive body fat condition, which leads to an increased risk for morbidity and/or premature mortality [1]. Childhood obesity has become a global epidemic and a serious public health challenge [2,3]. In Europe, data on childhood obesity are collected by the World Health Organization (WHO) European Childhood Obesity Surveillance Initiative (COSI), which was established in 2007 [4]. According to COSI, the prevalence of being overweight (including obese) was 29% in boys and 27% in girls aged 7–9 years; the prevalence of obesity was 13% in boys and 9% in girls [5]. Obesity negatively affects various body systems, leading to an increased risk of several health conditions, such as musculoskeletal conditions, cardiovascular and metabolic disorders, and gastrointestinal and respiratory diseases [3].

The association of obesity with musculoskeletal problems and incorrect body posture has been widely reported and could represent a causal factor for impairment of cardio–respiratory efficiency, degenerative bone processes, and back pain [6,7]. Body posture refers to the positioning of body segments and represents a critical determinant to preserve musculoskeletal health. It is important to prevent or to eliminate precociously excess fat mass in order to not only avoid cardiovascular and metabolic disease but also to prevent incorrect body posture and associated complications.

Healthy lifestyle strategies such as physical activity and exercise, diet, and behavioral changes are crucial players to manage and prevent obesity and obesity-related conditions in children and adolescents [8].

Regular physical exercise is essential for regular growth and development and should be considered a natural part of a healthy lifestyle from childhood [8]. Physical exercise is considered a non-pharmacological intervention that can reduce the health risks related to excess weight. In particular, exercise may also represent a tool to mitigate postural and musculoskeletal problems and to provide the somatosensory system with the information necessary to improve proprioception and representation of the body in space [9]. This rearrangement could lead to an improvement in the posture used during the performance of daily activities and could preserve the spine and joints [10]. Combined educational and therapeutic interventions may be useful to prevent weight gain and incorrect body posture.

In this narrative review, we aim to consider incorrect body posture in children and adolescents with obesity, focusing on the relationship with its negative impact on physical activity, and to discuss the role of exercise as a therapeutic approach. Appropriate early interventions for the management of musculoskeletal problems are needed to ensure healthy growth and to prevent comorbidities in childhood and adulthood.

## 2. Materials and Methods

We performed a narrative review, presenting a non-systematic analysis of the available literature on the topic of incorrect body posture in children and adolescents with obesity, its relationship with the negative impact on physical activity, and the role of exercise as a therapeutic approach. Original scientific papers, clinical trials, meta-analyses, and reviews of major relevance (Filter Child: 0–18 years) published in the past 15 years and articles in the English language were identified. Case reports or series and letters were not considered. The electronic databases PubMed, Scopus, EMBASE, and Web of Science were utilized for this research. The following keywords, alone and/or combined, were used: obesity, body posture, children, adolescents, physical activity, exercise, and musculoskeletal complications. The contributions were independently collected by V.R., A.P., R.G., P.P., G.S.S, A.I., and M.F., and reviewed and discussed by V.C, L.M., M.V., V.C.P., and V.F. The resulting draft was critically revised by V.C, L.M., M.V., and G.Z. The final version was then approved by all.

## 3. Childhood Obesity

Childhood obesity is one of the most severe public health problems of this century. According to a report from the Center for Disease Control (CDC) in the United States, in 2011–2014, the prevalence of obesity among children and adolescents aged 2–19 years was 18.9% [11]. In 2017–2021 that prevalence increased to 19.7% and affected about 14.7 million children and adolescents [12]. Childhood obesity and overweight are important risk factors for non-communicable diseases (NCDs) [12]. In this context, the “WHO Global Action Plan for the Prevention and Control of Non-communicable Diseases 2013–2025” set targets to halt the rise of obesity among young people [13], and the “WHO Comprehensive Implementation Plan on Maternal, Infant, and Young Child Nutrition” identifies as a priority stopping the increase of childhood overweight and obesity rates by 2025 [14]. Within the European context, the “WHO European Action Plan for the Prevention and Control of NCDs” recognizes the importance of prevention as the most viable option to reduce the obesity and overweight epidemic [15].

The most widely used method to measure and identify obesity is the Body Mass Index (BMI) [16]. According to CDC growth charts, in children aged 2 to 19, overweight is defined as a BMI above the 85th percentile for age and sex, whereas obesity is defined as a BMI above the 95th percentile [17].

In contrast, the WHO uses statistically based cut-offs corresponding to the number of standard deviations above the median. In children ages 0 to 5, the WHO uses a BMI > +2 SD to define overweight, which is equivalent to a BMI > 97th percentile, and a BMI > +3 SD to define obesity, which is equivalent to a BMI > 99th percentile. In children aged 5 to 19, a BMI > +1 SD defines overweight, and a BMI > +2 SD defines obesity [18].

### 3.1. Risk Factors

It is widely accepted that obesity results from an imbalance between energy intake and expenditure, with an increase in positive energy balance being closely associated with lifestyle and dietary intake preferences. However, there is increasing evidence indicating that an individual’s genetic background is important in determining obesity risk [2].

The “UK Foresight” report [19] presents a complex obesity system map influenced by over 100 variables acting at the individual, household, community, and wider societal levels. This multifactorial system can be conceptualized using Ecological Systems Theory (EST) [20]. According to EST, development or change in individual characteristics cannot be effectively explained without considering the context, or ecological niche, in which the person is embedded. In the case of a child, the ecological niche includes the family and the school, which are, in turn, embedded in larger social contexts, including the community and society in general. In addition to these larger contexts, specific characteristics of the child, such as gender and age, interact with familial and societal characteristics to influence growth. To summarize, according to EST, child characteristics interact with processes in the family and in the school, which themselves are influenced by characteristics of the community and society [20,21,22,23,24,25,26,27,28]; see Figure 1.

Genetics are one of the biggest factors in the cause of obesity [2]. Some studies have found that BMI is 25–40% heritable [21]. Obesity-related heritable traits are influenced by the interplay of genetics, epigenetics, metagenomics, and the environment. Forms of obesity resulting from mutations in a single gene affect less than 5% of the population.

### 3.2. Complications

Paediatric obesity is a multisystem condition associated with the initiation and maintenance of a chronic inflammatory status, which plays a crucial role in the development of inflammatory metabolic derangements.

Adipose tissue contains a wide variety of immune cells, which essentially turns it into an immunological organ [21,22,23]. Due to a chronic excess of nutrients and lipids, adipose tissue undergoes adaptive modifications aimed at satisfying metabolic needs. Parallel to the hypertrophy of adipocytes, there is a functional modification of the adipocytes characterized by an altered pattern of adipokine secretion [21]. This results in an increase in the expression and secretion of adipokines with pro-inflammatory action capable of inducing a low-grade state of inflammation; this involves neutrophil participation in the early phases, followed by macrophage involvement and mast cell polarization [20,22], and inducing the production of other pro-inflammatory cytokines, such as TNF-α. Moreover, it stimulates the production of IL-6, which, in turn, stimulates the production of acute-phase reactants such as C-reactive protein (CRP) [23]. Adipose-tissue-related inflammation, even in its early stages, leads to a wide variety of metabolic and cardiovascular complications.

As reported in Table 1, obesity-related consequences have short- and long-term complications, including metabolic–endocrine complications; cardiovascular complications, such as primarily endothelial dysfunction, hypertension, and atherosclerosis [2,24,25]; gastroenterological complications; respiratory complications; renal complications; neurological complications; and also dermatological complications [26,27,28,29,30,31,32,33,34]. In addition, paediatric obesity raises the risk of various musculoskeletal problems, impairing mobility, increasing fracture prevalence, and causing lower limb joint pain and malalignment [35]. Furthermore, it is also a risk factor for unilateral or bilateral slipped capital femoral epiphysis, tibia vara, valgus knee, and flat foot [36,37]. Lastly, psychological/psychiatric disorders, including anxiety, low self-esteem, depression, eating disorders, and social shame, are also frequent, affecting quality of life in adulthood and reducing life expectancy [2,27,38].

### 3.3. Treatment

Treatment of obesity in children and adolescents needs a multidisciplinary patient- and family-focused approach to achieve both short-term goals (e.g., decreasing food intake, lowering weight gain, improving body composition, increasing energy expenditure, and enhancing physical function) and long-term goals (e.g., reducing obesity-related comorbidities) [39,40]. According to the Childhood Obesity Task Force of the European Association for the Study of Obesity [41], treatment choice varies according to the severity of obesity, age, gender, pubertal status, psychosocial factors, possible comorbidities, patient and family preferences, and willingness to change.

First-line treatment of paediatric obesity relies on a nonpharmacological approach, which is based on implementing lifestyle interventions, including diet modification, increased physical activity, therapeutic exercise [7], and behavioural changes [42,43,44,45].

Moreover, psychological management of a child with obesity is a significant issue to consider, owing to the strong association between childhood obesity and mental disorders [46,47,48,49].

When this first-line approach is not sufficient, pharmacological treatment of obesity is employed as a second-line approach. It is considered for children with severe obesity and aged more than 10 years who have failed to respond to dietary and lifestyle treatments for one year, as well as those with multi-organ complications [50,51,52,53].

Eventually, adolescents with obesity who are non-responders to first- and second-line treatments may be offered metabolic and bariatric surgery [54].

## 4. Incorrect Body Posture in Children and Adolescents with Obesity

Body posture is subject to large changeability, which depends on many factors, such as sex, somatic type, age, ethnicity, psychophysical conditions, and environment [55], and it is a good predictor of present and future musculoskeletal health [56].

The assessment of body posture should constitute a crucial element of the complex examination of child health, in particular, in children with excessive body mass. In the literature, “healthy posture” is defined as: the state of muscular and skeletal balance which protects the supporting structures of the body against injury or progressive deformity [7].

According to some authors, the optimal posture involves mid-range position of the pelvis, slight lumbar lordosis, slight thoracic kyphosis, and with the head in a well-balanced position. With respect to the standing lateral view, the center of gravity should be anterior to the talus [57,58]. The ear, shoulder, hip, knee, and talus should be perfectly aligned; and posterior parts of the head, back, and gluteal muscles should be vertically aligned [59].

A correct body posture during childhood favorably affects the growth of the whole body: it contributes to normal development of organs and improves the efficiency of motor activity, which, in turn, contributes to normal development of muscles, joints, and ligaments and stimulates skeleton growth [55].

Postural impairments are one of the most common yet underestimated health issues during school age caused by human growth or pathological conditions [6]. If untreated, body posture impairments could cause a reduction in cardio–respiratory efficiency, worsening of bone and back pain, displacement of internal organs, degeneration of bone, and back pain [7,60]. These outcomes are predictors of several conditions, such as the thoracic kyphosis (a prominent curvature of the thoracic spine that creates a hunchback appearance), valgus lower limb (a lower leg deformity caused by deviation of the knee joint from the body’s mid-line), and lumbar hyper-lordosis (abnormal accentuation of the spine’s inward curvature in the lumbar region) [7,56,61]. Excessive body mass may decrease body stability and, consequently, cause postural mechanisms of adaptation, such as increased lumbar lordosis and pelvic anteversion, a rotation caused by forward projection of the iliac crest, increasing risk of falls.

Some studies have reported that the development of a child’s body posture impairments are related to the periods of the fastest pace of growth, which correspond to the beginning of school age (6–7 years) and puberty (12–16 years) [55].

Puberty is characterized by the increase of fat gain, body weight/BMI, and chest and shoulder dimensions. Girls, especially if tall, tend to slouch, positioning their shoulder incorrectly [7]. Children aged 13–15 years are reported to frequently have an asymmetric position of the shoulders, which is associated with overloads and carrying schoolbags on one side [62,63]. Excessive fat and lower levels of physical activity are often associated with long periods in a seated position during lessons and other school-related activities [62]. These behavioral patterns could promote different postural habits, such as more-pronounced thoracic kyphosis and downward head positioning [62,64]. This risk increases in children/adolescents with obesity and overweight [6,65,66] with a low level of physical activity and sedentary lifestyle. The most frequent alterations encountered in children with overweight and obesity are flat feet, head and shoulders in a protracted position, thoracic kyphosis, valgus lower limb, and lumbar hyper-lordosis [7,56,61]; see Figure 2.

Recently, a study by Macialcyzyk-Paprocka et al. [7] showed that 74% of children with an excessive BMI had transient and correctable deformities not caused by skeletal alterations but from vicious postural habits or pain, called paramorphism. This functional prognosis is easily reversible, especially if diagnosed early and treated in order to avoid structural bone modification, named dysmorphism [7,62]. Children with overweight or obesity are 1.5 times more likely to develop incorrect body posture than normal-weight children. This could be ascribed to non-optimal body segment alignment leading to muscle over-activation and mechanical stress [7,67]. Excessive lengthening or shortening, in combination with pelvic anteversion, leads to internal hip joint rotation, valgus knees, and flat feet [6,68].

The foot plays a fundamental role in movement and balance, and its development is influenced not only by internal factors such as sex, age, and genetics, but also by external factors such as shoes, physical activity, and weight. According to several studies, the tendency of children with obesity to suffer from flat feet could be significant [69].

Another study by Rusek et al. [55,70] investigated the association between BMI and postural vices and, contrary to the findings of Grabara et al., showed an increased distance between the scapula and the frontal plane in children with a higher BMI, which causes chest pain and dealignment from the midline (confidence level 95%; CI 0.05; *p* = 0.009) [55,62].

Recently, Bayartai et al., in a cross-sectional study, found that, in children, obesity was associated with increased thoracic kyphosis (95%; CI 10.10–15.80; *p* < 0.0001) and thoracic extension (95%; CI 2.90–11.60; *p* = 0.005), but decreased mobility in thoracic flexion (95%; CI 1.20–8.80; *p* = 0.01), lateral flexion (95%; CI 11.60–23.80; *p* < 0.0001), hip flexion and extension, lumbar flexion (95%; CI 8.70–15.50; *p* < 0.0001), extension (95%; CI 8.70–15.50; *p* < 0.0001), and lateral flexion (95%; CI 5.50–12.80; *p* < 0.0001) [71]. Reduced lumbar movement in overweight individuals could be explained by excess adipose tissue obstructing vertebral intersegmental mobility [72].

This association between overweight and thoracic kyphosis was also confirmed by Valdovino et al. in a retrospective comparative cohort study of 70 non-scoliotic adolescents and 1551 adolescents with idiopathic scoliosis. Overweight was associated with increased proximal thoracic kyphosis in both groups (T2–T5: *p* < 0.001; T5–T12: *p* < 0.001). According to the authors, excess fat may hinder anterior vertebral growth by increasing the compressive load on the vertebral growth plane [73].

However, according to Bayartai et al., as previously demonstrated by Shiri and Hartvigsen et al. in cross-sectional studies, lumbar and sacral posture were not different between children and adolescents with normal weight and obesity (*p* = 0.34) [71,74,75].

Excessive body weight identification and correction may ameliorate effectiveness of interventions and improve therapeutic action. In fact, previous studies have demonstrated that therapeutic exercise improved global joint mobility in young populations and reduced the prevalence of incorrect body posture in school-aged children. Finally, postural exercise has been demonstrated to increase performance and the ability to complete daily routine activities [6,7,37,38,39,40,41,42,43,44,45,46,47,48,49,50,51,52,53,54,55,56,57,58,59,60,61,62,63,64,65,66,67,68,69,70,71,72,73,74,75,76,77,78,79,80,81,82,83,84,85,86,87,88,89,90,91,92,93,94].

## 5. Impact of Body Posture on Physical Activity in Childhood Obesity

Physical effects of obesity are found mainly as impaired motor skills; in fact, excess weight has a strong impact on movement and imposes abnormal mechanics on the body that burden the musculoskeletal structures, which are forced to adapt to support the weight [76]. The shape of the body is therefore strongly influenced by the weight it has to support [77].

Generally, children with obesity have higher difficulties practicing sports and motor activities, which manifests as a reduction in conditional skills such as coordination, balance, running speed, agility, fine and gross motor skills, and hand–eye coordination [78,79,80,81,82,83,84,85]. In turn, having poor motor skills reduces motivation to participate in physical activity with peers, thus partly explaining a more sedentary lifestyle and establishing a “vicious circle” [86,87,88]. The major consequences to movement are important limitations when carrying out activities of daily living, such as walking [78]. This becomes considerably slower, the static phases increase, the steps shorten and tend to widen outwards to increase stability, and the step is also adapted in order to reduce the load on the knees and the metabolic expenditure of the gait [89]. Due to the displacement of the center of gravity and the weight to be supported, balance becomes precarious, with a consequent increase in the risk of falling [90]. Less physical activity is closely related to a reduction in muscle strength [3]. According to one study, this also derives from reduced function of the muscles from abnormal metabolism, and therefore from lower oxidative capacity of the muscle fibers [91]. In children with obesity, it is important to distinguish absolute strength from relative strength. The first is not affected, since obesity does not seem to have a negative effect on the intrinsic ability of the muscle to contract; however, excess fat mass and impaired motor coordination can be considered among the major causes of the reduced performance of these subjects, especially when the required task is directly proportional to mass—for example, when moving from a sitting to standing position [81,92,93,94,95].

Obesity, which is associated with the reduction in relative muscle strength, increases the risk of developing disabilities. There is a reduction in the ability of muscles and bones to absorb shock and a greater risk of injury, fractures, and osteoarthritis, especially in the knees [96]. Very often, these subjects also complain of increased pain in the neck, back, and lower limbs [97,98,99,100,101]. A high BMI significantly affects the spine, which is forced to bear excessive loads that generate functional overloads in the muscles and other structures of the spine [102,103], leading to incorrect curvature of the thoracic and/or lumbar spine, with consequent misalignments of the lower limbs [69,104,105,106].

A high BMI also has negative effects on energy expenditure during movement and cardiovascular response. It increases the rate of early fatigue and decreases the perceived fatigue level. Children with obesity are forced to consume a reduced amount of oxygen in relation to their mass and are likely to switch from aerobic to anaerobic exercise. Breathing can be compromised by the pressure of adipose tissue on the diaphragm [107]. These factors also lead children with obesity to adopt incorrect postures in order to support the activities of daily life.

Table 2 summarizes the studies that have investigated the association between weight and physical function.

Postural impairments can limit motor skills of children with obesity and, consequently, their ability to engage in physical activity and exercise. Therefore, there is the necessity to reduced barriers to exercise in order to promote correct development of the musculoskeletal system and to support physical activity practice for weight management. In this light, healthcare professionals should modify, adapt, and tailor exercise to optimize the efficacy of the therapeutic exercise.

## 6. Therapeutic Exercise Benefits for Incorrect Body Posture

During the period of growth, incorrect posture has a negative impact on children and adolescents with body weakness and the perception of physical defects; thus, appropriate use of physical exercise could be beneficial to restore correct posture [108]. In particular, therapeutic exercises prescribe specific movements to correct impairments, restore muscular and skeletal function, and/or maintain a state of well-being, with direct benefits for patients with incorrect body posture [109]. The role of exercise is two-fold and has indirect and direct effects. Exercise can improve poor body posture in those without obesity, but it is even more important for those with obesity because, along with directly improving body posture, it can help to reduce weight, which will, in turn, improve body posture. Moreover, exercise has a direct effect on the increase in body energy expenditure and the acquisition of motor skills, ameliorates of cardiovascular and metabolic profile, and contributes to reduce weight gain with positive effects on posture [109,110,111].

In fact, several studies have shown that with age, body posture issues more seriously compromise metabolism, including cardiopulmonary function and the skeletal system [110]. Indeed, previous studies have shown that many adolescents are sitting for long periods during school time, which leads to a muscular imbalance of their trunk muscles [111]. Lafond et al., in their postural study with children aged 4–12 years, described postural differences found in the sagittal plane that were influenced by the amount of time spent sitting, which increased postural translations in the sagittal plane [112]. A longitudinal study with children aged 11 years and followed for three years by Grivas et al. found a significant association between back pain and trunk asymmetry [113]. Hyperkyphosis is a typical paramorphism, but exercise interventions targeting back extensor muscle strength resulted in modest improvements in clinical measures of kyphosis [114]. Therefore, therapeutic exercise is an effective means to address paramorphisms, or alterations in body shape and attitude [111], particularly in subjects with obesity. This is crucial to avoid chronic changes that cannot be modified with corrective exercise that reduces muscle tension and works on correct muscle strengthening phases. These exercises aim to prevent the onset of possible deformities, ensuring healthy child development because the bones and joints become stabilized in a new trim [111]. In a review by Laita et al. [111], the duration of therapeutic programs had to be at least 8 weeks to obtain better results on postural control. The myofascial system is mainly responsible for correct postural maintenance, and through mechanical characteristics of connective tissue, it withstands and organizes the action of bones and muscles. In particular, most paramorphism are initially not visible and are painless, but tend to emerge during the school period. The literature indicates that over 50% of children from the first cycle of elementary school to the third year of middle school have paramorphisms affecting the spine, feet, and knees [115]. Latalski et al. showed how the lack of prophylaxis and neglecting adequate procedures may lead to limitations of physical and motor abilities, back pain, or the development of severe spinal deformities [116]. The creation of adequate conditions for the psychomotor development of children, as well as the elaboration and implementation of specific educational schemes tailored to schools and parents, is crucial to prevent physical alterations from turning into dysmorphisms [108]. The review by Han et al. (2018) concluded that visual, vestibular, and somatosensory systems, the systems contributing to postural maintenance, seemed to be affected by the excessive weight condition in children with obesity. Specifically, they found that the excessive pressure on the feet may modify plantar sensory receptor activity, which reduces the feedback required to adjust body position and maintain postural balance. Moreover, for the eyes-closed condition, children with obesity showed worsened body posture control when compared to their normal-weight peers. Furthermore, besides greater posture instability, children with obesity showed a higher time to correct their movements. Due to additional weight of the abdominal cavity, the position of the center of pressure is closer to the anterior edge of the base of support, and to maintain balance, it is necessary for them to adjust the torque of their ankles [117]. Thus, if the muscles are unable to respond quickly, the risk of falling is greater. The increased amplitude of motor commands leads to greater variability and an increased area of postural oscillation [118,119].

Exercise in children with scoliosis might lessen any potential reduction in physical function over time, and in general, physical fitness and exercise can improve children’s overall sense of well-being and happiness [120]. In fact, paramorphism correction and a change of incorrect attitudes can be obtained through constant and continuous stretching exercises of the myofascial retractions and strengthening of deficit muscles [121]. Ceballos Laita et al. confirmed the positive effects of corrective, therapeutic exercise in terms of reducing symptoms and improving function, as well as various angles and body asymmetries, in adolescent idiopathic scoliosis [111].

Collectively, studies have revealed the therapeutic potential of exercise in children with incorrect body posture in order to avoid serious postural impairments in the future. Therefore, therapeutic exercise should aim not only at correcting postural defects but also preventing possible future worsening. In fact, therapeutic exercise helps global and harmonic body development of children. Exercise programs should aim to prevent or correct posture impairment by focusing on reinforcing muscle strength and increasing proprioceptive activities, joint mobility, and stretching. In particular, muscle strength should focus on the reinforcement of main muscular groups such as the back, pectoral, abdominal, and lower limb muscles. To better work on postural stabilization, training should include exercises to implement body stability, such as quadrupedal position, with the use of small equipment such as elastic bands or medicine balls [122].

Wright et al. did not find improvements in fundamental movements when comparing two different intervention programs (i.e., a movement-based program vs. a generic multisport program) of 4-week duration [123]. The duration of the therapeutic exercise programs must be at least 8–12 weeks to obtain the ameliorant of the postural control. In fact, other studies included in our review have durations from 8 weeks to 6 months for exercise programs and find postural impairment improvement.

As previously reported, the crucial age for the development of postural impairments is during elementary school. For this reason, a combination of educational interventions in school settings that aim to prevent weight gain and to evaluate and correct postural impairments could help children and parents to ameliorate problems through prevention or correctional strategies. School campaigns alone cannot reverse the prevalence of childhood obesity—an ecosystem in which schools and families work together to create a healthy environment seems to be more effective [122].

In light of this, exercise programs can be considered a valuable tool for patients with incorrect posture, such as children with obesity, and the importance of correct posture should be emphasized as a prerequisite of a healthy life [7]. The exercise program must have a duration of at least of 8–12 weeks to produce positive effects on the postural outcomes [111]. There should be ongoing education for parents and teachers about postural and joint problems caused by obesity.

## 7. Conclusions

Obesity is associated with various dysfunctions of the organism, including incorrect body posture and musculoskeletal issues. The body reacts to excess weight by changing its normal balance, and the somatosensory systems of children with obesity are forced to make major adjustments to compensate for postural problems. For these reasons as well as others, sports and many types of general physical activity that require these adaptations are a serious issue for children with obesity. Thus, prevention programs must be based not only on the reduction of body weight but also on the definition of correct postural habits from an early age. Early evaluation of postural impairments could foster a tailored therapeutic exercise program. Parents and teachers should be informed about the possibility of postural impairments caused by obesity to better comply with treatments and to avoid long-term issues. It is equally important to provide correct information on the types and doses of physical activity that can help prevent these problems.

## Figures and Tables

**Figure 1 ijerph-19-16728-f001:**
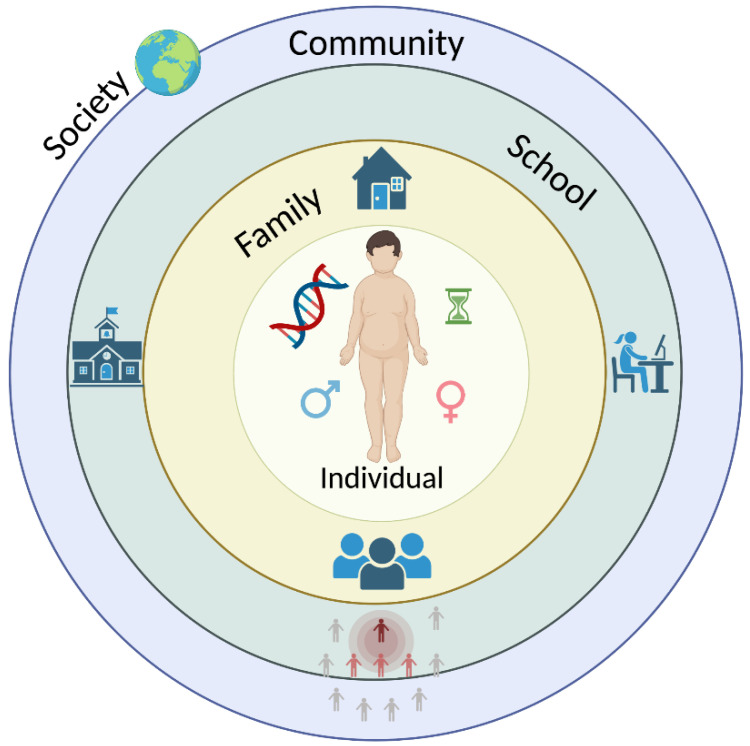
Multifactorial system involved in determining obesity risk.

**Figure 2 ijerph-19-16728-f002:**
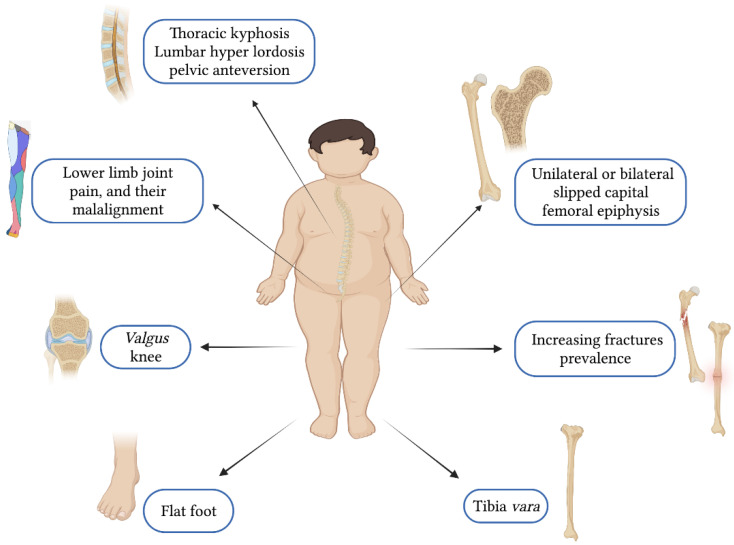
Body posture and musculoskeletal problems in children and adolescents with obesity.

**Table 1 ijerph-19-16728-t001:** Obesity-related short- and long-term complications.

System	Complications
Metabolic–endocrine	- Insulin resistance - Prediabetes - Diabetes mellitus - Dyslipidemia - Hyperandrogenism - Polycystic ovary syndrome (PCOS)
Cardiovascular	- Endothelial dysfunction - Hypertension - Atherosclerosis
Gastroenterological	- Non-alcoholic fatty liver disease (NAFLD) - Hepatic steatosis - Gastroesophageal reflux - Gallbladder disease
Respiratory	- Sleep obstructive disease - Asthma - Hypoventilation
Renal	- Focal diseases
Neurological	- *Pseudotumor cerebri* - Headache
Dermatological	- *Acanthosis nigricans* - *Striae rubrae* - Intertrigo - Hidradenitis suppurativa - Furunculosis

**Table 2 ijerph-19-16728-t002:** The literature exploring the association between weight and physical function.

Condition	Recent Key Supporting Evidence	Type of Study	Main Findings
Impaired coordination	Tsiros et al. [78] Barnett et al. [79]	UR SR	Motor skill development should be a key strategy in childhood interventions aiming to promote long-term physical activity.
Reduced motor skill proficiency	Tsiros et al. [78] Slotte et al. [85] Barnett et al. [79] Cattuzzo et al. [80] Mahaffey et al. [81]	UR SR SR SR SR	Motor skill development is necessary to promote motor competence in children and to enhance the participation in later motor activities such as sport-related or recreational activities.
Impaired balance (e.g., during challenging balance tasks involving a narrowed stance ± vision)	Tsiros et al. [78] Tsiros et al. [82] O’Malley et al. [83] Barnett et al. [79] Mahaffey et al. [81]	UR CS CS SR SR	Balance influences functional activities except sleeping. Poor balance could be ameliorated by working on coordination and propriocetive stimuli.
Reduced lower limb muscle strength (relative to body mass or during mass-dependent tasks)	Tsiros et al. [78] Rodrigues de Lima et al. [92] Garcia-Hermoso et al. [93] Grao-Cruces et al. [94] Mahaffey et al. [81] Thivel et al. [95]	UR SR SR SR SR SR	Children and adolescents with obesity display a decrease in muscular fitness compared to normal-weight peers.
Increased pain (e.g., musculoskeletal pain, neck/back pain, lower limb pain)	Tsiros et al. [78] Sanders et al. [98] Palmer et al. [99] Azabagic e Pranjic [100]	UR SR C C	The mantainment of normal weight from an early age reduces back pain in children.
Gait deviation (e.g., increased pelvic/hip/knee motion, prolonged stance phase, wider-based gait)	Molina-Garcia et al. [101]	SR	Children with obesity had higher difficulties, with greater step width, longer stance phase, higher tibiofemoral contact forces, higher ankle plantarflexion moments, and greater power generation.
Postural malalignment (increased lumbar lordosis, genu valgum)	Molina-Garcia et al. [102]	SR and MA	Children with obesity had a significantly higher risk of lumbar hyperlordosis, genu valgum, flatfoot, and any joint malalignment compared to their peers of normal weight.

C = cohort study, CS = cross−sectional study, SR = systematic review, MA = meta−analysis, UR = umbrella review.

## Data Availability

Not applicable.

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
