# Peer review of "Childhood Obesity and Incorrect Body Posture: Impact on Physical Activity and the Therapeutic Role of Exercise"

_ijerph, 2022, doi:10.3390/ijerph192416728_

Round 1

Reviewer 1 Report

Childhood obesity and incorrect body posture: impact on physical activity and the therapeutic role of exercise

Overview:

The authors present an interesting narrative review focusing on the impact of poor body posture in children and adolescents with overweight and obesity. The authors firstly describe childhood obesity in detail, including the definition, epidemiology, risk factors, complications and treatment of childhood obesity. The authors then present three key topics linked to body posture and obesity. These include how obesity can detrimentally affect body posture in children, how poor body posture can negatively impact physical activity, and how exercise can help to reduce poor body posture among children and adolescents. The authors draw a short conclusion bringing together the three topics. The authors make a good contribution to the literature by bringing together the evidence in these three areas. The authors bring the importance of body posture to light (an area which is often overlooked in the behavioural obesity literature).

General comments:

The research questions of the narrative review should be made clearer and more focused. The narrative review should focus on the association between body posture and obesity in children and adolescents, the impact of body posture on physical activity in childhood obesity, and the therapeutic benefits of exercise on body posture in children and adolescents with obesity.

The grammar and English language used throughout should be reviewed to improve readability, including reducing very long sentences.

The terminology of overweight and obesity should be reviewed throughout. ‘Children and adolescents with overweight / obesity’ should be used, rather than ‘overweight / obese children and adolescents'.

The authors should ensure all key scientific terms are defined throughout.

The childhood obesity section (section 3) could be reduced and incorporated into the introduction section to set the scene rather than being presented later on. This section could be edited to better link to and reflect the aims of the review and research questions. This is because the level of detail in some sections seem less relevant to the review aims. There should be clearer links made between the topics described in section 3 on childhood obesity and the review aims to improve flow between sections.

Greater critical appraisal of the literature throughout sections 4-6 could be used. The authors should make it clearer where the evidence base is stronger or weaker and make suggestions for future research throughout to inform the readers where the research gaps lie and where evidence may be inconsistent.

The flow between sections could be improved to improve readability.

Abstract:

Line 26-28: The review aims could be reworded for better clarity.

Line 30: Please change ‘be-come’ to ‘become’.

Line 32 – 33: The sentence could be reworded to better explain what these ‘serious problems’ may be.

Introduction: The introduction could present a reduced version of what is presented in section 3. The introduction could start with the definition of obesity; then briefly describe the epidemiology and risk factors of obesity; then briefly discuss the complications of obesity, highlighting the importance of changes to body posture; then briefly describe the possible treatment options for childhood obesity, highlighting the importance of exercise and how it can improve body posture and obesity.  

The definition of body posture and incorrect body posture should be included in the introduction.

Materials and Methods:

A materials and methods section is not necessarily required for a narrative review. If the authors choose to include this section, definition or age range of children and adolescents could be added.

Section 3: Childhood obesity

As mentioned in other comments, this section should be reduced and incorporated into the Introduction to set the scene.

The definition of obesity should be come before the epidemiology of obesity.

Lines 82 – 100: This section could be edited and reduced by referencing the data sources instead of describing them in full and by more concisely summarising the Global Action Plan, as this is less relevant to the aims of the review.

Lines 83 – 86: The authors should present the most up-to-date obesity statistics first, followed by how much this has risen in recent years.

Lines 117 – 159: This section on risk factors could be edited and reduced by more concisely summarising the Ecological Systems theory of obesity. A figure could be included to better illustrate the different factors associated with obesity.

Lines 134 – 137: The difference between genetics/genetic susceptibility/genetic factors should be better explained. On the one hand the authors state genetics are a big factor in obesity but then on the other state genetic factors account for less than 5% of obesity cases. A reference is also needed for the third statement in this paragraph.

151 – 159: It is not clear why you have specifically focused on describing poor eating habits and sedentary time as key factors associated with obesity here.

Lines 161 – 177: This section on complications seems less relevant to the research questions and so could be reduced and edited to be more of a lay summary. This section is also very biochemistry focused, whereas the rest of the article has a behavioural focus. For a reader with a behavioural focus on obesity, this section would be difficult to follow. Therefore, the level of detail may not be required.

Lines 166 – 167: This sentence should be reworded. I am not familiar with the term “immune organ”.

Line 177: What are these consequences and complications?

Lines 178 – 199: The grammatical structure of this paragraph could be reconsidered. I think it would read better if the authors started a new sentence for each ‘type of complication’. E.g. cardiovascular complications include… Pulmonary complications include… The authors could instead put this information in a table.

Line 192: The word lastly should be removed as the authors go onto to describe several more complications afterwards.

Lines 201 – 240: As the review is not focused on the treatment of obesity specifically, rather exercise as a treatment for poor body posture, this section is less relevant and could be reduced. The authors could mention that nonpharmacological and pharmacological treatment can be used to treat obesity, dependent on severity of obesity and gives examples of these treatments and when they may be offered. The specific drug names seem less relevant to this review. It would be useful to include statistics on how many children receive pharmacological and nonpharmacological treatment for obesity. It would also be useful to state whether therapeutic exercise to improve poor body posture is currently part of obesity treatment in children. This would then provide a better link into the next section.

Line 211: Please give an example of what you mean by behavioural changes.

Line 214: Please explain what ‘off-meals’ are.

Section 4: Incorrect body posture in children and adolescents with obesity

Line 242: Please remove the word represents

Line 267: Please define what you mean by ‘postural vices’

Line 270: Please define the terms used here - thoracic kyphosis, valgus lower limb, and lumbar hyper lordosis. Key terms could be introduced earlier on when introducing the topic of body posture.

Lines 280 – 281: Again, please define these key terms and ensure all key scientific terms are defined throughout.

Lines 288 – 291: Please expand on why ‘increased distance between the scapula and the frontal plane’ can be detrimental to children.

Lines 309 – 310: This sentence seems rather ad hoc and not very well linked to the previous section on body posture and obesity. The authors could include a short summary paragraph, summarising the literature presented above and suggesting what the literature means for the importance of obesity prevention and treatment in children.

Section 5: Impact of body posture on physical activity in childhood obesity

Line 317: I suggest using an alternative word to refusal. Do children with obesity refuse to practice sports or are they just less inclined or motivated to?

Line 319: I suggest rewording ‘ability with balls’ to ‘hand-eye coordination’

Lines 348 – 351: This section should be reworded to improve coherence. When ‘individuals’ are referred to in line 348, does this refer to individuals with obesity? Adolescents are mentioned in line 351, does this not refer to children as well?

Lines 352 – 253: This could be moved further up in the paragraph when physical function is described (i.e. around line 317).

Table 1: The title could be reworded to ‘Literature exploring the association between weight and physical function’. The acronyms used in the study type column need describing in a table footnote. The table is not overly informative, the overall findings from each study could be included.

A short summary section could be included at the end.

Section 6: Therapeutic exercise benefits for incorrect body posture

Lines 357-362: As this review is focused on obesity, this section and especially this first paragraph could also acknowledge the role of exercise in reducing obesity. The role of exercise is two-fold and has indirect and direct effects. Exercise can improve poor body posture in those without obesity but is even more important for those with obesity because it can help to reduce weight, which will in turn improve body posture as well directly improve body posture. I suggest this relationship between exercise, body posture and weight is described in more detail, otherwise the links between sections appear disjointed. Therapeutic exercise should be defined at the start of this sections and examples of therapeutic exercise should be given.

Line 307: The authors could include how long the children were followed up for in this longitudinal study

Line 371 - 373: Key terms should be defined.

Line 376: Please explain what corrective gymnastics is.

Line 378 - 380: I suggest not using the term ‘in fact’ throughout the article and rather use suggestive terms (e.g. literature suggests/indicates).

Lines 387 - 394: More detail on exercise programmes to prevent or correct poor body posture in children and adolescents could be added to this section. The authors could describe how these studies explored the role of therapeutic exercise in improving body posture. Where these studies experimental, lab-based? Have there been any school or home-based studies? If there has, these should be described and if there has not, the lack of evidence-based programmes should be acknowledged.

A short summary section could be included at the end.

Section 7: Conclusion

Line 403: Please remove ‘this section is mandatory’.

Line 408: References are not required in the conclusion. The conclusion should be the authors summary of the literature they have reviewed, rather than a summary of others work.

Line 412: Here the authors mention the importance of parents and teachers, however, there has been little mention of their important in the previous sections. Authors should avoid introducing new ideas in the conclusion and instead summarise their work. However, I do agree that education and understanding of parents and teachers is important and therefore should be included in section 6.

The conclusion could include specific suggestions for future research, which would enhance this field.

Author Response

Reviewer 1

Overview:

The authors present an interesting narrative review focusing on the impact of poor body posture in children and adolescents with overweight and obesity. The authors firstly describe childhood obesity in detail, including the definition, epidemiology, risk factors, complications and treatment of childhood obesity. The authors then present three key topics linked to body posture and obesity. These include how obesity can detrimentally affect body posture in children, how poor body posture can negatively impact physical activity, and how exercise can help to reduce poor body posture among children and adolescents. The authors draw a short conclusion bringing together the three topics. The authors make a good contribution to the literature by bringing together the evidence in these three areas. The authors bring the importance of body posture to light (an area which is often overlooked in the behavioral obesity literature).

R: we thanks the reviewer for their appreciation and their corrections that surely improved our work.

General comments:

The research questions of the narrative review should be made clearer and more focused. The narrative review should focus on the association between body posture and obesity in children and adolescents, the impact of body posture on physical activity in childhood obesity, and the therapeutic benefits of exercise on body posture in children and adolescents with obesity.

The grammar and English language used throughout should be reviewed to improve readability, including reducing very long sentences.

The terminology of overweight and obesity should be reviewed throughout. ‘Children and adolescents with overweight / obesity’ should be used, rather than ‘overweight / obese children and adolescents'.

The authors should ensure all key scientific terms are defined throughout.

The childhood obesity section (section 3) could be reduced and incorporated into the introduction section to set the scene rather than being presented later on. This section could be edited to better link to and reflect the aims of the review and research questions. This is because the level of detail in some sections seem less relevant to the review aims. There should be clearer links made between the topics described in section 3 on childhood obesity and the review aims to improve flow between sections.

Greater critical appraisal of the literature throughout sections 4-6 could be used. The authors should make it clearer where the evidence base is stronger or weaker and make suggestions for future research throughout to inform the readers where the research gaps lie and where evidence may be inconsistent.

The flow between sections could be improved to improve readability.

Abstract:

Line 26-28: The review aims could be reworded for better clarity.

Line 30: Please change ‘be-come’ to ‘become’.

Line 32 – 33: The sentence could be reworded to better explain what these ‘serious problems’ may be.

R: we revised the abstract (page 1)

Introduction: The introduction could present a reduced version of what is presented in section 3. The introduction could start with the definition of obesity; then briefly describe the epidemiology and risk factors of obesity; then briefly discuss the complications of obesity, highlighting the importance of changes to body posture; then briefly describe the possible treatment options for childhood obesity, highlighting the importance of exercise and how it can improve body posture and obesity.  

The definition of body posture and incorrect body posture should be included in the introduction.

R: we revised the introduction, including the definition and highlighting the topic of the review (pages 1-2). However we preferred not to remove the section 3, that we revised and shortened

Materials and Methods:

A materials and methods section is not necessarily required for a narrative review. If the authors choose to include this section, definition or age range of children and adolescents could be added.

R: thank you for your comment. According to “Gregory, A.T.; Denniss, A.R. An Introduction to Writing Narrative and Systematic Reviews — Tasks, Tips and Traps for Aspiring Authors. Heart Lung Circ. 2018, 27, 893–898, doi:10.1016/j.hlc.2018.03.027” a methodology also for narrative review could be useful. For this reason, we added the details of the age, without to remove the section.

Section 3: Childhood obesity

As mentioned in other comments, this section should be reduced and incorporated into the Introduction to set the scene.

The definition of obesity should be come before the epidemiology of obesity.

Lines 82 – 100: This section could be edited and reduced by referencing the data sources instead of describing them in full and by more concisely summarising the Global Action Plan, as this is less relevant to the aims of the review.

Lines 83 – 86: The authors should present the most up-to-date obesity statistics first, followed by how much this has risen in recent years.

Lines 117 – 159: This section on risk factors could be edited and reduced by more concisely summarising the Ecological Systems theory of obesity. A figure could be included to better illustrate the different factors associated with obesity.

Lines 134 – 137: The difference between genetics/genetic susceptibility/genetic factors should be better explained. On the one hand the authors state genetics are a big factor in obesity but then on the other state genetic factors account for less than 5% of obesity cases. A reference is also needed for the third statement in this paragraph.

151 – 159: It is not clear why you have specifically focused on describing poor eating habits and sedentary time as key factors associated with obesity here.

Lines 161 – 177: This section on complications seems less relevant to the research questions and so could be reduced and edited to be more of a lay summary. This section is also very biochemistry focused, whereas the rest of the article has a behavioural focus. For a reader with a behavioural focus on obesity, this section would be difficult to follow. Therefore, the level of detail may not be required.

Lines 166 – 167: This sentence should be reworded. I am not familiar with the term “immune organ”.

Line 177: What are these consequences and complications?

Lines 178 – 199: The grammatical structure of this paragraph could be reconsidered. I think it would read better if the authors started a new sentence for each ‘type of complication’. E.g. cardiovascular complications include… Pulmonary complications include… The authors could instead put this information in a table.

Line 192: The word lastly should be removed as the authors go onto to describe several more complications afterwards.

Lines 201 – 240: As the review is not focused on the treatment of obesity specifically, rather exercise as a treatment for poor body posture, this section is less relevant and could be reduced. The authors could mention that nonpharmacological and pharmacological treatment can be used to treat obesity, dependent on severity of obesity and gives examples of these treatments and when they may be offered. The specific drug names seem less relevant to this review. It would be useful to include statistics on how many children receive pharmacological and nonpharmacological treatment for obesity. It would also be useful to state whether therapeutic exercise to improve poor body posture is currently part of obesity treatment in children. This would then provide a better link into the next section.

Line 211: Please give an example of what you mean by behavioural changes.

Line 214: Please explain what ‘off-meals’ are.

R: thank you for your suggestion. We revised the section accordingly to the suggestions (pages 3-5). The section was shortened and same sentences were removed; figure 1 and table 1 were added. We reworded some sentence (before submission the manuscript has undergone English language editing by MDPI, ID 52911). We detailed the therapeutic role of exercise on incorrect body posture in the section 6.

Section 4: Incorrect body posture in children and adolescents with obesity

Line 242: Please remove the word represents

Line 267: Please define what you mean by ‘postural vices’

R: Thanks for your suggestion, we removed it and better-defined postural vices.

Line 270: Please define the terms used here - thoracic kyphosis, valgus lower limb, and lumbar hyper lordosis. Key terms could be introduced earlier on when introducing the topic of body posture.

Lines 280 – 281: Again, please define these key terms and ensure all key scientific terms are defined throughout.

Lines 288 – 291: Please expand on why ‘increased distance between the scapula and the frontal plane’ can be detrimental to children.

R: Thanks for your suggestion, we corrected the lines.

Lines 309 – 310: This sentence seems rather ad hoc and not very well linked to the previous section on body posture and obesity. The authors could include a short summary paragraph, summarising the literature presented above and suggesting what the literature means for the importance of obesity prevention and treatment in children.

R: Thanks for your suggestion, we revised it.

Section 5: Impact of body posture on physical activity in childhood obesity

Line 317: I suggest using an alternative word to refusal. Do children with obesity refuse to practice sports or are they just less inclined or motivated to?

Line 319: I suggest rewording ‘ability with balls’ to ‘hand-eye coordination’

R: Thanks for your suggestion, we revised them.

Lines 348 – 351: This section should be reworded to improve coherence. When ‘individuals’ are referred to in line 348, does this refer to individuals with obesity? Adolescents are mentioned in line 351, does this not refer to children as well?

R: Thanks for your suggestion, we corrected it.

Lines 352 – 253: This could be moved further up in the paragraph when physical function is described (i.e. around line 317).

Table 1: The title could be reworded to ‘Literature exploring the association between weight and physical function’. The acronyms used in the study type column need describing in a table footnote. The table is not overly informative, the overall findings from each study could be included.

A short summary section could be included at the end.

R: Thanks for your suggestion, we corrected all.

Section 6: Therapeutic exercise benefits for incorrect body posture

R: we revised the section 6 (pages 10-12)

Lines 357-362: As this review is focused on obesity, this section and especially this first paragraph could also acknowledge the role of exercise in reducing obesity. The role of exercise is two-fold and has indirect and direct effects. Exercise can improve poor body posture in those without obesity but is even more important for those with obesity because it can help to reduce weight, which will in turn improve body posture as well directly improve body posture. I suggest this relationship between exercise, body posture and weight is described in more detail, otherwise the links between sections appear disjointed. Therapeutic exercise should be defined at the start of this sections and examples of therapeutic exercise should be given.

R: Thanks for your suggestion, we appreciated it and corrected in the text.

Line 307: The authors could include how long the children were followed up for in this longitudinal study

R: we added it.

Line 371 - 373: Key terms should be defined.

Line 376: Please explain what corrective gymnastics is.

Line 378 - 380: I suggest not using the term ‘in fact’ throughout the article and rather use suggestive terms (e.g. literature suggests/indicates).

R: Thanks for your suggestion, we appreciated it and corrected in the text.

Lines 387 - 394: More detail on exercise programmes to prevent or correct poor body posture in children and adolescents could be added to this section. The authors could describe how these studies explored the role of therapeutic exercise in improving body posture. Where these studies experimental, lab-based? Have there been any school or home-based studies? If there has, these should be described and if there has not, the lack of evidence-based programmes should be acknowledged.

A short summary section could be included at the end.

R: Thanks for your suggestion, we added this information in the text.

Section 7: Conclusion

Line 403: Please remove ‘this section is mandatory’.

Line 408: References are not required in the conclusion. The conclusion should be the authors summary of the literature they have reviewed, rather than a summary of others work.

R: Thanks for your suggestions, we corrected it.

Line 412: Here the authors mention the importance of parents and teachers, however, there has been little mention of their important in the previous sections. Authors should avoid introducing new ideas in the conclusion and instead summarise their work. However, I do agree that education and understanding of parents and teachers is important and therefore should be included in section 6.

The conclusion could include specific suggestions for future research, which would enhance this field.

R: Thanks for your suggestion, we added all (page 12).

Reviewer 2 Report

This narrative review is clearly structured and written. The authors explained the prevalence of childhood obesity and the incorrect body posture that caused by the obesity. A table showed the association between weight and phyiscal function revealed the impact that childhood obesity had on phyiscal activity. In the end, the authors discussed the therapeutic exercise benefits for correcting the body posture. Up to date, there is few review on this topic. This review focus in discussing the assocation of incorrect body posture and physical activity, and how therapeutic exercise would correct the body posture. To gain insight into the body posture improvements, it is important to discuss the baseline characteristics of body posture such as Craniocervical angle (Ëš), Thoracic flexion (Ëš), Trunk angle (Ëš), Plumb-tragus distance (cm), etc. 

Furtheremore, to address the therapeutic exercise benefits for incorrect body posture or obeisty, I will suggest the authors list a table listing a summary of reviews and meta-analysis for exercise interventions for weigh loss or body posture correction, and also list the outcome of the intervention if that's available. By doing this, the review will be more clear and provide more information regarding the therapeutic exercise benefits.

On top of that, the duration of excersise should also be discussed. Since Wright et al. did not find improvements in fundamental movements when comparing 2 different intervention programs (i.e., a movement-based program vs. a generic multisport program) of 4-week duration. (Wright MD, Portas MD, Evans VJ, Weston M. The effectiveness of 4 Weeks of fundamental movement training on functional movement screen and physiological performance in physically active children. J Strength Cond Res 29: 254–261, 2015. )

The authors published a narrative review discussing medical treatment of weight loss in children with obesity on 2022, where they discuss thoroughly the most up-to-date evidence on medical treatment of weight loss in children and adolescents with obesity, including FDA- or EMA-approved and -experimented, not approved, drugs for pediatric population. I will suggest the authors considering refining 3.5.Treatment part and give more space to the section 6.

Author Response

Reviewer 2

This narrative review is clearly structured and written. The authors explained the prevalence of childhood obesity and the incorrect body posture that caused by the obesity. A table showed the association between weight and phyiscal function revealed the impact that childhood obesity had on phyiscal activity. In the end, the authors discussed the therapeutic exercise benefits for correcting the body posture. Up to date, there is few review on this topic. This review focus in discussing the association of incorrect body posture and physical activity, and how therapeutic exercise would correct the body posture. To gain insight into the body posture improvements, it is important to discuss the baseline characteristics of body posture such as Craniocervical angle (Ëš), Thoracic flexion (Ëš), Trunk angle (Ëš), Plumb-tragus distance (cm), etc. 

R: Thanks for your suggestion, we added it in the section 4. 

Furtheremore, to address the therapeutic exercise benefits for incorrect body posture or obesity, I will suggest the authors list a table listing a summary of reviews and meta-analysis for exercise interventions for weigh loss or body posture correction, and also list the outcome of the intervention if that's available. By doing this, the review will be clearer and provide more information regarding the therapeutic exercise benefits.

R: Thanks for your suggestions. We agreed with your opinion but, to the best of our knowledge, we find only 1-2 review/meta-analysis on body posture correction (main topic of the manuscript), so we included the information in the text without providing a table.

On top of that, the duration of excersise should also be discussed. Since Wright et al. did not find improvements in fundamental movements when comparing 2 different intervention programs (i.e., a movement-based program vs. a generic multisport program) of 4-week duration. (Wright MD, Portas MD, Evans VJ, Weston M. The effectiveness of 4 Weeks of fundamental movement training on functional movement screen and physiological performance in physically active children. J Strength Cond Res 29: 254–261, 2015. )

R: Thanks, we added this information in the section 6

The authors published a narrative review discussing medical treatment of weight loss in children with obesity on 2022, where they discuss thoroughly the most up-to-date evidence on medical treatment of weight loss in children and adolescents with obesity, including FDA- or EMA-approved and -experimented, not approved, drugs for pediatric population. I will suggest the authors considering refining 3.5.Treatment part and give more space to the section 6.

R: we revised and shortened the section 3 and revised section 6

Round 2

Reviewer 1 Report

The authors have taken on board comments and made improvements to the manuscript. However, I suggest the manuscript requires additional English language editing and I have made several suggestions to improve readability below. 

Abstract 

lines 32-35: sentence starting "children with obesity had worst body control" requires rewording to improve English language. The sentence should perhaps read 'Children with obesity have less body control and functional ability due to the excess fat mass, which reduces their ability to perform motor skills and take part in physical activity.'

Introduction

line 44: a reference for the definition of obesity is required

line 50: please replace 'It negatively affects' to 'Obesity negatively affects'

lines 56-57: you write 'body posture represents refers...' - please remove represents

line 64: you may want to consider not abbreviating physical exercise to PE because in the UK and other countries this is a common abbreviation for Physical Education taught in schools. Also the abbreviation is only used in this one paragraph and not throughout the manuscript. 

line 74: please remove 'the' so the sentence reads 'negative impact on physical activity' to improve readability. 

Section 3. Childhood obesity

lines 145-146: The sentence starting 'heritable trait influenced by...' is not very clear, please reword. Perhaps to 'Obesity-related heritable traits are influenced by the interplay of genetics, epigenetics, metagenomics and the environment. Forms of obesity resulting from mutations in a single gene affect less than 5% of the population."

line 250: this does not need to be its own paragraph. I suggest adding it to the end of the last paragraph. 

Section 4. Incorrect body posture

line 260: please change 'children health' to 'child health'

line 274-276: please reword to 'a decrease in lung vital capacity' and 'a worsening of bone and back pain' to improve readability. 

line 277: please add 'the' before 'thoracic spine.

lines 279-280: please revise the placement of brackets in this sentence - you have 2 closed brackets without any open brackets.

line 281: please changes 'causes' to 'cause' 

line 283: please change 'with higher risk of falls' to 'increasing risk of falls' to improve readability. 

line 288: please change 'shoulders' to 'shoulder'

line 295: please change 'overweigh' to 'overweight' 

line 304: please remove 'a' before the word 'transient' - this is a plural sentence. 

line 306: please change to 'diagnosed early' 

lines 308-309: I suggest rewording to 'than normal weight children', The word classmate seems a bit informal in this context. 

line 309: please remove 'a' and reword to 'non-optimal body segment alignment' - make sure plural and non-plural sentences are structured correctly.

line 321: please change to 'children with obesity'

line 339-343: Good to finish section with a summary but this sentence requires English language editing to improve readability, as it is currently not overly clear. 

Section 5. Impact of body posture 

 line 350: please remove 'in' before the word 'children' and change 'had' to 'have'.

lines 351-353: you may want to remove coordination as you then go onto mention coordination as a conditional skill so the sentence reads 'reduction in conditional skills, such as coordination...' 

lines 362-364: you state 'according to the studies analysed..' but then only provide one study reference. Please either change to 'according to one study' or add additional references. 

lines 381-382: please reword to 'children with obesity 'are' forced... their mass 'and' are likely...'

Table 1 - third column: please change '...recreational ones' to 'recreational activities'  

Table 1 - fourth column (balance): balance is spelt incorrectly. The sentence starting 'poor balance' is not very clear and needs editing to improve readability. 

Table 1 - fifth column: please change to 'a decrease in' 

Line 394: please change maximized to 'reduce' - I suggest you want to reduce barriers to exercise not maximise them?

Section 6. Therapeutic exercise

lines 405-411: this section requires references to support your points. 

line 419: please change to 'longitudinal study with children aged 11 years, followed for three years,...' to improve readability. 

line 427: comma is required after the word deformities. I suggest changing to 'ensuring healthy child development' 

line 428: please change 'review of' to 'review by'

line 430: please change to 'is mainly responsible for correct...'

line 432: please change to organizes 

line 446: please change to 'eyes-closed condition, children' 

line 461: This sentence requires more context and a reference. 

lines 463-464: muscle strength should not be plural, please change throughout this paragraph.

lines 472-474: this sentence requires editing to improve readability. I suggest you reduce the number of times 'in fact' is used throughout the manuscript. 

line 476: remove 'the' before primary school. In a previous section earlier you referred to the USA education system. I suggest you keep this consistent throughout the manuscript - e.g. primary school is referred to in the UK but in the USA is referred to as elementary school. 

line 477: please reword to 'school settings, which aim to prevent...'

line 480: please change to 'work' - incorrect use of plural

line 484: please change to exercise programs 

line 485: I suggest the word guarantee is too strong

Conclusion

lines 496-499: This sentence requires editing to improve readability, as it is not very clear.

Author Response

Reviewer 1

The authors have taken on board comments and made improvements to the manuscript. However, I suggest the manuscript requires additional English language editing and I have made several suggestions to improve readability below. 

R: Thanks for your suggestions, we improved it.

Abstract 

lines 32-35: sentence starting "children with obesity had worst body control" requires rewording to improve English language. The sentence should perhaps read 'Children with obesity have less body control and functional ability due to the excess fat mass, which reduces their ability to perform motor skills and take part in physical activity.'

R: we corrected it.

Introduction

line 44: a reference for the definition of obesity is required

R: we added the reference

line 50: please replace 'It negatively affects' to 'Obesity negatively affects'

lines 56-57: you write 'body posture represents refers...' - please remove represents

line 64: you may want to consider not abbreviating physical exercise to PE because in the UK and other countries this is a common abbreviation for Physical Education taught in schools. Also the abbreviation is only used in this one paragraph and not throughout the manuscript. 

line 74: please remove 'the' so the sentence reads 'negative impact on physical activity' to improve readability. 

R: we corrected the sentences.

Section 3. Childhood obesity

lines 145-146: The sentence starting 'heritable trait influenced by...' is not very clear, please reword. Perhaps to 'Obesity-related heritable traits are influenced by the interplay of genetics, epigenetics, metagenomics and the environment. Forms of obesity resulting from mutations in a single gene affect less than 5% of the population."

line 250: this does not need to be its own paragraph. I suggest adding it to the end of the last paragraph. 

R: Thanks for your improvements, we appreciated.

Section 4. Incorrect body posture

line 260: please change 'children health' to 'child health'

line 274-276: please reword to 'a decrease in lung vital capacity' and 'a worsening of bone and back pain' to improve readability. 

line 277: please add 'the' before 'thoracic spine.

lines 279-280: please revise the placement of brackets in this sentence - you have 2 closed brackets without any open brackets.

line 281: please changes 'causes' to 'cause' 

line 283: please change 'with higher risk of falls' to 'increasing risk of falls' to improve readability. 

line 288: please change 'shoulders' to 'shoulder'

line 295: please change 'overweigh' to 'overweight' 

line 304: please remove 'a' before the word 'transient' - this is a plural sentence. 

line 306: please change to 'diagnosed early' 

lines 308-309: I suggest rewording to 'than normal weight children', The word classmate seems a bit informal in this context. 

line 309: please remove 'a' and reword to 'non-optimal body segment alignment' - make sure plural and non-plural sentences are structured correctly.

line 321: please change to 'children with obesity'

line 339-343: Good to finish section with a summary but this sentence requires English language editing to improve readability, as it is currently not overly clear. 

R: Thanks for your suggestions, we improved the sentences.

Section 5. Impact of body posture 

 line 350: please remove 'in' before the word 'children' and change 'had' to 'have'.

lines 351-353: you may want to remove coordination as you then go onto mention coordination as a conditional skill so the sentence reads 'reduction in conditional skills, such as coordination...' 

lines 362-364: you state 'according to the studies analysed..' but then only provide one study reference. Please either change to 'according to one study' or add additional references. 

lines 381-382: please reword to 'children with obesity 'are' forced... their mass 'and' are likely...'

Table 1 - third column: please change '...recreational ones' to 'recreational activities'  

Table 1 - fourth column (balance): balance is spelt incorrectly. The sentence starting 'poor balance' is not very clear and needs editing to improve readability. 

Table 1 - fifth column: please change to 'a decrease in' 

Line 394: please change maximized to 'reduce' - I suggest you want to reduce barriers to exercise not maximise them?

R: Thanks for your suggestions, we improved the sentences.

Section 6. Therapeutic exercise

lines 405-411: this section requires references to support your points. 

R: we added the references

line 419: please change to 'longitudinal study with children aged 11 years, followed for three years,...' to improve readability. 

line 427: comma is required after the word deformities. I suggest changing to 'ensuring healthy child development' 

line 428: please change 'review of' to 'review by'

line 430: please change to 'is mainly responsible for correct...'

line 432: please change to organizes 

line 446: please change to 'eyes-closed condition, children' 

line 461: This sentence requires more context and a reference. 

lines 463-464: muscle strength should not be plural, please change throughout this paragraph.

lines 472-474: this sentence requires editing to improve readability. I suggest you reduce the number of times 'in fact' is used throughout the manuscript. 

line 476: remove 'the' before primary school. In a previous section earlier you referred to the USA education system. I suggest you keep this consistent throughout the manuscript - e.g. primary school is referred to in the UK but in the USA is referred to as elementary school. 

line 477: please reword to 'school settings, which aim to prevent...'

line 480: please change to 'work' - incorrect use of plural

line 484: please change to exercise programs 

line 485: I suggest the word guarantee is too strong

R: Thanks for your suggestions, we improved the sentences.

Conclusion

lines 496-499: This sentence requires editing to improve readability, as it is not very clear.

R: Thanks for your suggestions, we improved the sentence.
